# OpenReview forum: "Causes and Consequences of Representational Similarity in Machine Learning Models"
_ICLR.cc/2026/Conference — Submitted to ICLR 2026_

### Official Review · Reviewer_dQv7 · 2025-10-31

**Soundness:** 2
**Presentation:** 3
**Contribution:** 1
**Rating:** 2
**Confidence:** 4

**Summary:**

This paper investigates the causes of high representational similarity among machine learning models across both vision and language domains.
The authors hypothesize that dataset overlap and task overlap are the main drivers of this phenomenon.
They introduce dataset and task splitting techniques to systematically control these overlaps and evaluate their effects across multiple architectures, including ResNets, ViTs, diffusion models, and large language models.
Empirical results show that greater overlap leads to higher representational similarity, and that models with more similar representations are more susceptible to transferable adversarial and jailbreak attacks.

**Strengths:**

[S1] The paper addresses an interesting and timely question related to model alignment and generalization.

[S2] The experimental framework is well-designed and systematic, covering multiple model types and data modalities.

**Weaknesses:**

[W1] Limited novelty of findings. The main results of this paper, that overlapping datasets or tasks yield more similar representations and higher attack transferability, are intuitive and have already been shown in prior literature.
For example, [1] treats training dataset differences as a key cause of representational dissimilarity, and [2] explores model similarity through adversarial attack transferability across different architectures and training regimes. Similar patterns have also been reported in [3], which links network similarity directly to attack transferability.
Thus, the contributions of this paper are largely confirmatory and show low novelty beyond well-understood phenomena.

[1] What shapes feature representations? Exploring datasets, architectures, and training (NeurIPS 2020)
[2] Similarity of Neural Architectures using Adversarial Attack Transferability (ECCV 2024)
[3] The Relationship Between Network Similarity and Transferability of Adversarial Attacks (2025)

[W2] Limited technical depth and scope. The proposed dataset and task splitting methods are straightforward and mainly procedural rather than methodological innovations. The experiments rely on small-scale synthetic datasets and LoRA-tuned LLMs, which limit the generalizability of the conclusions to large-scale or real-world model training. In addition, the study uses a single similarity metric (CKA) and simple regression fitting, resulting in limited technical rigor and diversity in evaluation.

**Questions:**

- How does this work provide new insights beyond prior studies [1–3] that already link dataset/task overlap and attack transferability to representational similarity? (see W1)

- Since the study relies mainly on a single similarity metric (CKA) and simple regression analysis, how can the authors ensure that the observed trends are robust across different similarity measures or evaluation methods? (see W2)

---

> ### Author Response · Authors · 2025-11-23
> **Initial response (1)**
>
> We thank the reviewer for the review. We appreciate the reviewer for finding the question we explore interesting & timely, and for acknowledging our paper's systematic and well-designed experiments. In the updated PDF (**all updates are in blue** and longer new content is noted with text like "new ablation" or "new discussion" in blue), we
>
> 1. provide additional ablation on model size, architecture, and tuning duration in section A.2.4; specifically:
>     - Figure 19 investigates the effects of tuning duration,
>     - Figure 20 explores how model size ratio between pairs of models affects representational similarity at a fixed overlap,
>     - Figures 21 and 22 fix the model pairing (ResNet18+ResNet152 in Figure 21, ResNet50+ViT/Small in Figure 22) while varying the overlap;
> 2. modify our claims in the conclusion, introduction, and abstract to correspond more closely to our results;
> 3. add previously-missing trend lines to plots in the appendix;
> 4. include additional discussion of related works in section A.5;
> 5. discuss nanoGPT results in more detail (section A.2.2);
> 6. present full-model tuning (Figure 24) of Llama3.2-1B-Instruct in addition to LoRA tuning.
>
>
> We respond to the reviewer's questions and concerns below.
> > [W1][Q1] Limited novelty of findings.
>
> We appreciate the reviewer highlighting [1–3]. We refer to [1] in our related work section in the main body. These papers are relevant for our submission, but they address different questions than those addressed by our submission. Our work does not focus on "confirming" that similar models are susceptible to transfer attacks (we simply demonstrate one downstream effect of representational similarity as an auxiliary experiment to provide credence to our claims regarding downstream effects in the introduction); our work is about **quantifying** how two tuning-time factors (namely, dataset and task overlap) contribute to representational similarity in the first place. We secondarily explore one concrete downstream implication (transferrable attack) of model similarity.
>
>
> Prior work either examines how features evolve within a single model under different tasks [1], or correlates similarity with transferability across pre-existing models trained on the same dataset [2,3]. We agree that our section titled "Downstream Consequences of Representational Similarity" has overlap with [2,3]; however our main focus is disentangling the effects of dataset and task overlap, instead of investigating the downstream susceptibility to transfer attacks. None of these papers manipulate the degree of overlap between the tuning datasets of two independently tuned models. In contrast, we construct controlled dataset- and task-overlap scenarios and quantify how representational similarity changes due to changing overlap.
>
> [1] study feature selection within a single model: which features (shape, texture, color) are represented under different training tasks, and how trained models compare to untrained ones. They do not tune pairs of models on systematically overlapping/disjoint datasets and do not vary dataset overlap between models. The comparisons in [1] across tasks are about which features are encoded by a model, not about how dataset overlap between independently trained models causally controls representational similarity across those models.
>
> Authors in [2] define a similarity metric (SAT) based on adversarial attack transfer and apply the metric to many ImageNet classifiers that are all trained on the same dataset and task. They explicitly state that their goal is to “quantify the similarity between general NNs,” focusing on architectural components, not training data/task overlap.
>
> Similarly, [3] develops a representational similarity metric (DBS); they additionally evaluate many pre-trained CNNs on transfer attacks, but again treats pretrained networks as is; they do not manipulate the tuning procedure of networks in order to isolate effects on representational similarity or susceptibility to transfer attacks. Their analysis is post hoc over an existing zoo of models.
>
> We also disentangle dataset overlap from task overlap using a purpose-built dataset that keeps inputs fixed while varying labeling schemes. This allows us to show that task overlap alone can drive similarity, and that combined dataset+task overlap yields the strongest predictive relationship. This separation is absent from [1–3], which do not provide mechanisms for independently varying data and task alignment between models. Although [1] constructs synthetic datasets with direct relationships between features and labels, similar to our `ColorShapeDigit800K` dataset, they focus their exploration on what causes features to be suppressed/learnt.

---

> > ### Author Response · Authors · 2025-11-23
> > **Initial response (2)**
> >
> > > [W1][Q1][Response cont'd]
> >
> > Finally, our analysis spans multiple modalities--including image classification, image generation, and language models--and we connect overlap-induced similarity to downstream transfer-attack success. Prior work [2,3] studies attack transfer as a similarity metric, but only in settings where models share both dataset and task. Our results demonstrate that overlap in training distributions, not just architecture, is a key driver of both representational alignment and transferability.
> >
> > Please refer to the section titled "Additional discussion of related work" in the updated PDF for discussion within the paper.
> >
> >
> > >[W2][Q2] Limited technical depth and scope.
> >
> > We respectfully disagree that the work lacks technical depth. The contribution is not the mechanics of splitting datasets but the *causal isolation* of dataset- and task-overlap effects, which prior studies have not been able to quantify. In reading the literature on representational similarity, we noticed that numerous papers measure representational similarity in various ways, few works have investigated actual causes of this phenomenon. Most papers seem to take it for granted that representational similarity occurs. We found this reliance on intuition unsatisfactory, and it motivated our investigation. We believe that there is value in revisiting fundamental questions like this to ensure that our understanding of ML model behavior is well-investigated. Driven by this motivation, we designed experiments that isolated two variables (task & data overlap) so that we can investigate them independently (e.g., with the ColorShapeDigit800K dataset) as well as investigate their joint effects. ColorShapeDigit800K dataset—enables varying task overlap while holding data constant, a condition unavailable in prior work such as [4,5]. This design provides methodological novelty by allowing one to measure the independent and joint contributions of task and data overlap, and our paper’s results show that these factors are strong, quantifiable drivers of representational similarity.
> >
> >
> > > The experiments rely on small-scale synthetic datasets
> >
> > We believe there are still lessons to be learned from the smaller-scale experiments presented in this paper. Working with smaller models allows us to control the entire training from end to end so that we can quantify overlap. While the difference in scale between the models we explore and today’s massive LLMs may cause unexpected side effects, we believe our results provide some intuitive understanding of what could happen in larger models. Additionally, numerous works in the adversarial machine learning space (e.g. Carlini, N., & Wagner, D. (2017). Towards Evaluating the Robustness of Neural Networks, among many others) highlight the dangers of "transferable adversarial attacks," in which malicious inputs constructed to fool one model can fool another one with similar representations. Our work which aims to understand causes of such similarities could enable the construction of better defenses against such transfer attacks.
> >
> >
> > The evaluation is also broader than suggested. The paper reports results across multiple modalities and architectures, including extended experiments on large ResNets and ViTs, which were previously added to address concerns about scale. These experiments show positive correlations between overlap and representational similarity across modalities (with effects being the strongest for vision tasks), providing empirical foundations for understanding how these effects may manifest in larger systems.
> >
> >
> > > LoRA-tuned LLMs
> >
> > In Figure 24, we provide new experimental results on continued pretraining of full LLMs, which tunes the full model instead of a low-rank adapter. Results of LoRA-tuned and full-tuned LLMs are similar.
> >
> >
> > > limit the generalizability of the conclusions to large-scale or real-world model training
> >
> > Without pretraining large models similar to how large corporations pretrain chatbots, it is difficult to say with certainty if conclusions generalise to real-world model pretraining. Due to limited compute, we are unable to produce pretraining experiments of large models with a large amount of data and focus on tuning pretrained large models (specifically, Llama).
> >
> > We respectfully disagree that results on LoRA-tuning or full-tuning of LLMs cannot be generalised to real-world use of large models. Real-world machine learning researchers/practitioners in both industry (with the exception of the most resourceful corporations) and academia are heavily dependent on tuning pretrained large models, since training from scratch is prohibitively expensive and challenging.
> >
> > Additionally, our work follows prior literature in using these smaller models and synthetic datasets. The reviewer referred to [1,2,3], which all use relatively simple DNNs, and they all evaluate their methods on small, synthetic datasets.

---

> > > ### Author Response · Authors · 2025-11-23
> > > **Initial response (3)**
> > >
> > > >the study uses a single similarity metric
> > >
> > > The main body of the paper does in fact present CKA as the only metric. However, **our appendix presents full results on seven representational similarity metrics** (including CKA), and we disagree with the statement that we rely on a single similarity metric. In many cases, we see broad agreement between different similarity metrics.
> > >
> > > >simple regression fitting
> > >
> > > We argue that regression analysis, although simple, is appropriate for our purposes, which measures the correlation between two scalar variables (overlap and representational similarity or overlap and attack success rate). For example, authors in [1] also present results using linear fits (linear regression), which we find appropriate for the setting. Other methods of demonstrating trends between two scalar variables include Pearson/Spearman correlation (note that Pearson correlation is equivalent to $R^2$ in linear regression with intercept), polynomial regression, and generalised additive models. Different correlation metrics will likely show very similar conclusions as linear regression & $R^2$ scores, and more complicated fitting methods would be unsuitable for a simple statistical analysis task involving scalar independent and dependent variables.
> > >
> > > In addition to regression analysis, we present mutual information between overlap and representational similarity in Tables 1 and 6. In aggregate, Table 1 supports our main conclusion that dataset overlap is the weakest driver for representational similarity. Therefore, we disagree with the claim that we rely inappropriately on regression analysis.
> > >
> > >
> > >
> > >
> > > [1] What shapes feature representations? Exploring datasets, architectures, and training (NeurIPS 2020)
> > >
> > > [2] Similarity of Neural Architectures using Adversarial Attack Transferability (ECCV 2024)
> > >
> > > [3] The Relationship Between Network Similarity and Transferability of Adversarial Attacks (2025)
> > >
> > > [4] Training objective drives the consistency of representational similarity across datasets, 2024.
> > >
> > > [5] The platonic representation hypothesis, ICML 2024.
> > >
> > >
> > >
> > > We thank the reviewer for their time. Please feel free to respond with any questions/requests for clarifications.

---

### Official Review · Reviewer_k4Mo · 2025-10-31

**Soundness:** 3
**Presentation:** 2
**Contribution:** 3
**Rating:** 6
**Confidence:** 3

**Summary:**

Recent work hypothesizes that as AI models scale up and train on more tasks, their representations will converge because their task sets increasingly overlap. This paper empirically tests what drives representational similarity by isolating two factors: dataset overlap (shared training datapoints) and task overlap (shared objectives/classes). The authors develop controlled splitting methods to vary these factors independently and train model pairs (ResNets, ViTs, LLMs, diffusion models) with different overlap levels, measuring representational similarity via CKA. They introduce ColorShapeDigit800K, a vision dataset enabling task variation while keeping images identical. For vision models, the results are clear: both dataset and task overlap increase representational similarity, with combined overlap showing the strongest effect. Language model results (Llama fine-tuning with LoRA) are largely inconclusive, showing weak or absent correlations—whether this reflects LoRA limitations or fundamental differences remains unclear. The authors explore downstream consequences by testing vulnerability to transfer attacks: models with higher representational similarity show increased susceptibility to adversarial examples (vision), suggesting that unintended similarity between models may create shared security vulnerabilities.

The work is rigorous and provides valuable empirical evidence for the vision domain, though language results and restriction to same-architecture-and-size comparisons can be limiting.

**Strengths:**

- Clearly motivated work addressing an important gap: trying to understand *causes* rather than just observing representational similarity
- Well-designed methodology to isolate dataset and task overlap effects through controlled splitting
- Broad empirical evaluation across multiple modalities (vision, language, diffusion)
- Creative ColorShapeDigit800K dataset enabling task manipulation without dataset variation
- Novel connection to downstream security implications (adversarial transferability)
- Extensive appendix with multiple similarity metrics and detailed experimental settings
- Well written and enjoyable to read!

**Weaknesses:**

- Language model results are inconclusive and undermine the paper's central claims of dataset and task overlap strongly influencing the representational similarity:
    - Figure 19 (Llama fine-tuning): Only small positive correlations appear with CKA for task splitting; data splitting and local similarity measures show no effect from increasing overlap. Authors attribute this to LoRA without further hypothesis. Can the authors hypothesize beyond "LoRA limitations"? Is this a fundamental difference in how language vs vision models respond to fine-tuning, or purely a methodological artifact?
    - NanoGPT (trained from scratch): No linear trends provided, making assessment difficult
    - Critical: Since Llama results show no positive correlation with overlap increase across most metrics, the abstract/conclusion claims about "both task and dataset overlap" driving similarity might be overstated. The evidence supports this claim for vision models but not language models. The paper could either remove language claims or explicitly limit conclusions to vision models.
- Data vs Task vs Data+Task overlap analysis unfairly compares vision and language models when task overlap metrics only apply to vision (Section 5.3).
- As mentioned in the discussion, the work is limited to same-architecture comparisons; making the same observations cross-architecture results (ResNet vs ViT or small vs large architectures) would strengthen claims about the influencing factors of representational alignment.
- Figure 4:
    - ResNet101 shows anomalous high variance compared to other ResNets (Figure 4).  Also visible when using other similarity metrics.
    - Limited discussion of why ViTs show surprisingly low similarity values despite similar training protocols to ResNets. I assume that the ViTs have also been pretrained with ImageNet. Comparing two models after finetuning on TinyImageNet, shouldn’t they have much higher CKA values?
    - Could the authors comment on these?
- Some figures raise questions about correlation validity (e.g., Figure 6's high R² when points cluster at extremes). Can the authors comment on this?
- Presentation of key results could be improved: The paper uses boxplots with regression lines throughout, which becomes repetitive across dozens of panels with small fonts. While 7 similarity metrics are computed (appendix), the main text focuses on CKA minimally explaining this choice or what different metrics reveal. Better aggregation (summary tables, effect size comparisons across conditions) and/or diverse (aggregated) visualization approaches would significantly improve clarity.
- Missing minor details: which pretrained weights were used? Assumed ImageNet.

**Questions:**

- Please see questions already stated in weaknesses :)
- Could you provide cross-architecture comparisons (ResNet18 vs ResNet152, ResNet vs ViT)? While same-architecture controls for confounds, demonstrating the trend across architectures would significantly strengthen the claim that dataset/task overlap drives similarity generally.
- Figure 4: ResNet101 shows much larger variance than ResNet18/50/152. What causes this outlier behavior? Why do other ResNets show such tight distributions?
- Figure 4: How many CKA values are in one box?
- CKA values for ViTs fall in the 0.4-0.6 range where the metric has known interpretability issues (high variability in this region, Figure 4). How confident are you in the regression lines fitted to these mid-range values? Would bootstrapping over data subsets provide more reliable similarity estimates?
- Figure 6: The high R² values seem driven by clustering at extremes rather than a true linear relationship. The level 2 combinations appear to form a blob in the middle. Can you explain this pattern? Why aggregate all level 2 combinations rather than separating them?
- Section 5.2 conclusion (2) states task splitting influences representation alignment "independent of data-overlap" at 100% data overlap. Wouldn't showing results at 50% data overlap make this argument stronger?
- The task overlap metric only applies to vision models, making Section 5.3 comparisons across modalities questionable. Can acknowledge this limitation more explicitly?
- Could you elaborate on why you focus primarily on CKA when 7 metrics were computed? What do the other metrics reveal that CKA (with a linear kernel) doesn't? CKA can also be used with different kernels, allowing to measure similarity more locally.

---

> ### Author Response · Authors · 2025-11-23
> **Initial response (1)**
>
> We thank the reviewer for the thorough and detailed review. We are glad that the reviewer found many strengths in this submission and found it an enjoyable read. In the updated PDF (**all updates are in blue** and longer new content is noted with text like "new ablation" or "new discussion" in blue), we
>
> 1. provide additional ablation on model size, architecture, and tuning duration in section A.2.4; specifically:
>     - Figure 19 investigates the effects of tuning duration,
>     - Figure 20 explores how model size ratio between pairs of models affects representational similarity at a fixed overlap,
>     - Figures 21 and 22 fix the model pairing (ResNet18+ResNet152 in Figure 21, ResNet50+ViT/Small in Figure 22) while varying the overlap;
> 2. modify our claims in the conclusion, introduction, and abstract to correspond more closely to our results;
> 3. add previously-missing trend lines to plots in the appendix;
> 4. include additional discussion of related works in section A.5;
> 5. discuss nanoGPT results in more detail (section A.2.2);
> 6. present full-model tuning (Figure 24) of Llama3.2-1B-Instruct in addition to LoRA tuning.
>
>
> We respond to the reviewer's questions and concerns below.
>
>
> > [W1] Language model results are inconclusive
>
> Thank you for pointing this out. We have modified the abstract, introduction, experimental results, and discussion to reflect the weaker language model results. We would like to point out that the tuned Llama models exhibit moderate correlation between task overlap and representational similarity (see Figures 5, 24). However, we concede that the transfer jailbreaking for Llama together with dataset overlap results for Llama and nanoGPT indicate that dataset overlap is not a strong driver for representational similarity in language models.
>
>
> > [W1a] Figure 19 (Llama fine-tuning): Only small positive correlations appear with CKA for task splitting
>
> Note that this figure has been re-numbered to Figure 23. Additionally, we present results for full-tuned Llama models in Figure 24.
>
> > [W1a] data splitting and local similarity measures show no effect from increasing overlap. ... Can the authors hypothesize beyond "LoRA limitations"?
>
> In short: language datasets are larger and contain more general information, leading to models being more general after being trained on split datasets.
>
>
> A larger dataset causes each split to be more general, meaning that it is more likely for each dataset to contain enough varied data to train models that are better at generalising. We used TinyStories and Shakespeare for dataset overlap on nanoGPT, and both contain around 2 million tokens. In contrast, the largest classical image dataset we use is TinyImageNet, containing 100k images, with TinyStories containing 20 times more tokens than TinyImageNet has images. We admit this is a simplistic way of measuring the information contained within a dataset.
>
> We note that the above hypothesis is supported by diffusion model results: they exhibit weak correlation between overlap and representational similarity. Although we trained each diffusion model on CIFAR-10, the score matching training procedure increases the number of unique inputs $x_t\sim N(x_0,t^2 I)$ beyond the 50k images present in CIFAR-10 due to adding random Gaussian noise to all images. Hence, this provides evidence that training with more information reduces the effect of overlap on similarity.
>
> The reviewer might wonder if the results in Figure 19 (finetuning duration ablation for ResNet-50 on CIFAR-100) contradict the above hypothesis; it does not. Since we repeat over many epochs, there is no new data introduced.
>
>
> > [W1b] NanoGPT (trained from scratch): No linear trends provided
>
> Thanks for pointing this out. We update our figures with linear trends for nanoGPT and ResNet-18 results in the appendix (Figures 9, 10, 13, 14).
>
> > [W1c] claims about "both task and dataset overlap" driving similarity might be overstated
>
> Thank you for pointing this out. We have removed/qualified our conclusions and contributions. Please see updated text in the abstract, introduction, experimental results, and discussion which address this concern.
>
> > [W2,Q7] Data vs Task vs Data+Task overlap analysis unfairly compares vision and language models when task overlap metrics only apply to vision (Section 5.3)... The task overlap metric only applies to vision models, making Section 5.3 comparisons across modalities questionable.
>
> We agree that Section 5.3 presents an unclear comparison. We edited Table 3 to explicitly acknowledge that Task Overlap is for ResNet/ViT only and modified our conclusions. Please refer to Table 6 for disaggregated results for each model architecture for a fairer comparison. Due to the 9-page limit throughout rebuttal, we keep the two-row table in Section 5.3.

---

> > ### Author Response · Authors · 2025-11-23
> > **Initial response (2)**
> >
> > > [W3,Q1] ... limited to same-architecture comparisons. ...  cross-architecture results ... would strengthen claims ...
> >
> > Thank you for the suggestions. Please refer to Figures 20,21,22 for the requested ablation on cross-architecture comparisons. Additionally, Figure 19 investigates the effects of tuning duration.
> >
> > To summarise, we find that when the two models in a pair have different sizes/architectures, the effect of dataset overlap is diminished, whereas task overlap remains a strong driver of similarity; when tuning vision models for longer, there is a stronger correlation between dataset overlap and similarity, whereas the effects of task overlap is slightly diminished when tuning duration increases. Please refer to the discussion in section A.2.4 for additional details.
> >
> >
> > > [W4a,Q2] Figure 4: ResNet101 shows anomalous high variance
> >
> > We finetune all the ResNet and ViT models using the same three scripts (task overlap, dataset overlap, and `ColorShapeDigit800K`) and hyperparameters. The high variance may be due to unfortunate choices for the random seed or training-related hyperparameters.
> >
> > > [W4-Q3] Figure 4: ... How many CKA values are in one box?
> >
> > There are four per box in Figure 4. Please refer to Section A.1.1 for full descriptions of the number of random seeds, which may vary by experiment.
> >
> >
> > > [W4b,Q4] Figure 4: ... Limited discussion of why ViTs show surprisingly low similarity values despite similar training protocols to ResNets
> >
> > This may be due to a quirk of certain representational similarity metrics. In the appendix (Figure 17, ResNet/ViT with dataset splitting) we present full results for all similarity metrics. We note that the two metrics Edit-kNN and CKNNA have absolute values in roughly the same range for both ViTs and ResNets. Even within the same architecture the range of metric values may vary from model to model (e.g., first row of Figure 17: ResNet-18 ranges between 0.93-0.95, ResNet-152 between 0.78-0.83). We hope that the additional metrics we provide addresses this issue, since each metric behaves differently.
> >
> > Additionally, we surmise architectural differences between ResNets and ViTs may be a **confounding factor** for representational similarity metrics. Although we did not find any papers directly pointing to model architecture being a confounding factor, [1] indicates that the structure of input data as *one* confounding factor may lead to CKA being uncorrelated with functional behaviour of models. Architecture may be another confounding factor, especially in the indirect cross-architecture comparison that the reviewer makes in their comment. The submission's focus is on the relative trend in similarity metrics rather than the absolute values, so our conclusions are not affected by different absolute values of CKA metric for different models.
> >
> > As is common in existing work, we compute the representational similarity metrics for all models on a deterministic subset of the validation dataset.
> >
> >
> > >[W5] Some figures raise questions about correlation validity...
> >
> > Linear regression analysis is not a perfect measure for correlation, since the underlying data may not be linear. Therefore, we provide mutual information as another metric (see Tables 1,6).
> >
> > > [Q5] Figure 6: The high R^2 values seem driven by clustering at extremes rather than a true linear relationship. The level 2 combinations appear to form a blob
> >
> > We agree that the clustering at extremes and the large blob in the middle suggests that the underlying relationship between overlap and similarity may be nonlinear. However, for the sake of consistency and due to the small number of distinct x-axis values (only three distinct overlap levels in total), we did not explore a different fitting strategy. In Figure 11, we plot disaggregated task overlap (i.e., without the blob) for ResNet-18 trained from scratch.  There are three different possible pairings for overlap level 2 where one task overlaps: `'shape_color+digit_color'==2C, 'shape_digit'+'digit_color'==2B, 'shape_digit'+'shape_color'==2A`. Since the overlap at level 2 is fixed at 50% regardless of the pairing, we lump them together for the sake of presentation. ResNet/ViT finetuning results should look similar.

---

> > > ### Author Response · Authors · 2025-11-23
> > > **Initial response (3)**
> > >
> > > > [Q6] Section 5.2 conclusion (2) states task splitting influences representation alignment "independent of data-overlap" at 100% data overlap. Wouldn't showing results at 50% data overlap make this argument stronger?
> > >
> > >
> > > We agree that ablating on data overlap for our synthetic dataset would strengthen our claims. We settled on 100% as the dataset overlap for section 5.2 conclusion (2) because we intended to mean with fixed data overlap when we wrote "independent of data overlap"; additionally, 100% dataset overlap is convenient for running experiments.
> > >
> > > The `ColorShapeDigit800K` dataset enables us to change task overlap between levels 1, 2, and 3 at (almost) arbitrary dataset overlap as shown in the toy example below.
> > >
> > > Say that we have two colors, two shapes, and two digits, and the dataset has 24 images where each label triplet (x,y,z) contains three images (e.g., Color0Shape0Digit0 would contain 3 images). Each image is uniquely identified by the label triplet and the index of that image for a given label triplet, represented as `(x,y,z-n)` for the n-th image having label triplet (x,y,z). To satisfy the 50% data overlap requirement, we have in one split these images `(0,0,0-a), (0,0,0-b), (0,0,1-a), (0,0,1-b) ... (1,1,1-a), (1,1,1-b)`, and in another split these images `(0,0,0-a), (0,0,0-c), (0,0,1-a), (0,0,1-c) ... (1,1,1-a), (1,1,1-c)`. To have 50% task overlap, we can simply re-label all images in each split to the appropriate label.
> > >
> > > We decided to prioritise other ablations.
> > >
> > >
> > > >[W6a] Presentation of key results could be improved.
> > >
> > > Thank you for the suggestions, it is indeed difficult to parse the small text in the plots, and we do rely on boxplots as the main method of visualisation. However, we would like to defer updates to presentation in a future draft due to (a) the short window for rebuttal and (b) the authors prioritising additional experimental results.
> > >
> > > > [W6b]  the main text focuses on CKA minimally explaining this choice or what different metrics reveal.
> > >
> > > We agree we should've explained this in more detail. Please see the first paragraph of Section A.2 for a slightly more in-depth treatment. Due to space limitations, we defer discussion to the appendix. The main reasons for choosing different metrics are that (a) we refer to a well-regarded paper [2] to choose the metrics we use and (b) different metrics behave differently [3]. Other works such as [3] explore the differences between various similarity metrics in great detail, so we refer to other work for a more in-depth treatment.
> > >
> > > > [Q8] Could you elaborate on why you focus primarily on CKA when 7 metrics were computed? What do the other metrics reveal that CKA (with a linear kernel) doesn't? CKA can also be used with different kernels, allowing to measure similarity more locally.
> > >
> > > CKA is a reliable metric [4], and the authors of CKA [5] demonstrated its many reliable properties (e.g., CKA with a linear kernel satisfies desirable invariances to orthogonal transformations and scaling). However, [2] pointed out in section A the limitations of CKA, which "has a very strict definition of alignment". Therefore, authors in [2] use CKNNA which is a modified version of CKA as well as other metrics. According to Figure 11 in [2], the main difference between the metrics is that some are global (CKA, SVCCA) whereas others are local ($k$-NN based methods, in our case we use $k=10$ for local comparisons).
> > >
> > > Our goal is to demonstrate how dataset/task overlap may affect representational similarity and not to pick the best metric for our experiments; since different metrics each have their strengths and weaknesses, we decided to use a wide range of metrics found in [2].
> > >
> > >
> > > > [W7] Missing minor details: which pretrained weights were used? Assumed ImageNet.
> > >
> > > You are correct. We include this information in Section A.1.1.
> > >
> > >
> > > [1] Deconfounded Representation Similarity for Comparison of Neural Networks, NeurIPS 2022
> > >
> > > [2] The Platonic Representation Hypothesis, ICML 2024
> > >
> > > [3] Similarity of Neural Network Models: A Survey of Functional and Representational Measures, ACM Computing Surveys
> > >
> > > [4] Reliability of CKA as a Similarity Measure in Deep Learning, ICLR 2023
> > >
> > > [5] Similarity of Neural Network Representations Revisited, ICML 2019
> > >
> > > We thank the reviewer for their time and dedication, and for producing a very thoughtful & thorough review. Please feel free to respond with any questions/requests for clarifications.

---

> ### Author Response · Authors · 2025-12-02
> **Additional Experiments for ColorShapeDigit800K**
>
> >[Q6] Section 5.2 conclusion (2) states task splitting influences representation alignment "independent of data-overlap" at 100% data overlap. Wouldn't showing results at 50% data overlap make this argument stronger?
>
> We did not fully address this comment in our original rebuttal response since we did not run experiments *in time for our initial response*; **these experiments are now complete**. As stated in our initial response during the rebuttal period, we agree that using 50% (or even 0%) dataset overlap would make our argument stronger. We are grateful for the suggestion made by reviewers.
>
> For the sake of completeness, we ran task overlap method (2) experiments using 0%, 25%, 50%, and 75% data overlap on the `ColorShapeDigit800K` dataset to verify that the effects of task overlap persists regardless of dataset overlap. Please refer to Figure 29 in the updated PDF for the relevant plots.

---

### Official Review · Reviewer_NmnH · 2025-10-31

**Soundness:** 3
**Presentation:** 3
**Contribution:** 2
**Rating:** 4
**Confidence:** 3

**Summary:**

The authors systematically study how the two different factors (task and data overlap) contribute to the representational similarity of models trained from scratch. They created a new dataset called ColorShapeDigit800K which maintains a fixed set of data points while varying the task definition. With this dataset, they show that both task and dataset overlap cause higher representational similarity and that combining them provides the strongest effect. They further show that a high similarity between model representations also increases the vulnerability to transferable adversarial and jailbreak attacks between the one model to the other.

**Strengths:**

* The paper studies an interesting question that can have large implications for building and evaluating foundation models.

* The paper is well written and the experiments and results are described clearly.

* The authors make an effort to make the underlying factors configurable and do a good job of holding one property constant (e.g. data overlap) while varying the other (e.g. task overlap).

**Weaknesses:**

* The different tasks that are studied for ColorShapeDigit800K are all simple character/digit/color classification tasks while the difference between different SSL objectives (multi-view, masking), image/text, supervised models, text models seem to be larger in my view. I’m not sure if it is possible to approximate this with these simple tasks.

* The results that more tasks and data overlap increase representationals similarities feel a bit trivial. Although it is nice that the authors confirm that in a controlled way, I’m missing a novel insight from these results.

* It would be interesting to investigate whether these are the only factors that contribute to representational similarity or whether other factors (architecture, model size, training duration) also have a significant impact.

**Questions:**

* How were the representations exactly extracted for the different models?

* Why didn’t you use the CKNNA measure from the The Platonic Representation Hypothesis when you already point out downsides of CKA in the Limitations section?

---

> ### Author Response · Authors · 2025-11-23
> **Initial response (1)**
>
> We thank the reviewer for the review. We are glad that the reviewer found the question we investigate interesting and the paper well-written. In the updated PDF (**all updates are in blue** and longer new content is noted with text like "new ablation" or "new discussion" in blue), we
>
> 1. provide additional ablation on model size, architecture, and tuning duration in section A.2.4; specifically:
>     - Figure 19 investigates the effects of tuning duration,
>     - Figure 20 explores how model size ratio between pairs of models affects representational similarity at a fixed overlap,
>     - Figures 21 and 22 fix the model pairing (ResNet18+ResNet152 in Figure 21, ResNet50+ViT/Small in Figure 22) while varying the overlap;
> 2. modify our claims in the conclusion, introduction, and abstract to correspond more closely to our results;
> 3. add previously-missing trend lines to plots in the appendix;
> 4. include additional discussion of related works in section A.5;
> 5. discuss nanoGPT results in more detail (section A.2.2);
> 6. present full-model tuning (Figure 24) of Llama3.2-1B-Instruct in addition to LoRA tuning.
>
>
> We respond to the reviewer's questions and concerns below.
>
>
> > [W1] The different tasks that are studied for ColorShapeDigit800K are all simple character/digit/color classification tasks while the difference between different SSL objectives (multi-view, masking), image/text, supervised models, text models seem to be larger in my view. I’m not sure if it is possible to approximate this with these simple tasks.
>
>
> We use a different "presentation" (definition) of task overlap: instead of task being defined as the training objective or loss function (self-supervised, supervised image classification, image segmentation etc), we have defined it as what the model is trained to do, which is heavily dependent on training data. Therefore, using our definition it is the case that task overlap has not been investigated in the context of representational similarity. The contribution of training objective to representational similarity has been investigated extensively [1]. We do not define task overlap to strictly be related to the training objective. Hence, our analysis and experiments are novel and provide evidence for the intuitive assumption that data and task overlap drive representational similarity.
>
> We chose this particular definition of task overlap (what the model is designed to do) because we can more easily quantify the task overlap. It is difficult to quantify how much "overlap" there is between different SSL objectives, making a quantitative analysis (which we provide in the paper using our own definition of task) impossible. Given two models trained on two different SSL objectives, there is no meaningful metric that we can use to quantify the similarity (or overlap) between these two objectives. In order to measure how much (i.e., quantify) the task overlap affects the representational similarity, we used our definition of task overlap, making it quantifiable. Concretely, 2 models trained to classify color+shape and color+digit, respectively, have a 50% task overlap (the color is the overlap).
>
> Based on our definitions of task overlap (i.e., what the model is designed to do), we note that task and dataset overlap are intimately related and it is difficult to design experiments where dataset and task overlap are chosen arbitrarily and independently. This motivated our construction of the `ColorDigitShape800k` dataset. Although simple, `ColorDigitShape800k` allows us to quantify how much overlap there is.
>
> We note that [2] also uses relatively simple synthetic datasets to investigate representational similarity.

---

> ### Author Response · Authors · 2025-11-23
> **Initial response (2)**
>
> > [W2] The results that more tasks and data overlap increase representationals similarities feel a bit trivial. Although it is nice that the authors confirm that in a controlled way, I’m missing a novel insight from these results.
>
>
> We thank the reviewer for this feedback and would like to provide some additional context on why we wrote this paper. In reading the literature on representational similarity, we noticed that numerous papers measure representational similarity in various ways, very few works have investigated actual causes of this phenomenon. Most papers seem to take it for granted that representational similarity occurs. We found this reliance on intuition unsatisfactory, and it motivated our investigation. We believe that there is value in revisiting fundamental questions like this to ensure that our understanding of ML model behaviour is well-grounded. Driven by this motivation, we designed experiments that isolated two variables (task & data overlap) so that we can investigate them independently (e.g., with the ColorShapeDigit800K dataset) as well as investigate their joint effects.
>
> While we agree that the core hypothesis is intuitive, we actually believe our hypothesis and results on vision models are somewhat surprising based on prior work! Authors in [3] observed that models trained on distinct datasets/modalities show signs of representational alignment. This would seem to directly contradict our observations on smaller models and finetuning larger models, as the models tested in [3] share neither dataset nor task, and could imply that some other factor (e.g. universal "truths" in datasets, common algorithmic pathways due to optimization functions) causes well-studied representational similarities between larger foundational models. Therefore, our findings that dataset and task similarity drive representational alignment provides interesting empirical evidence (caveat: we investigate small-ish models or tuning, rather than pretraining foundation models) of a hypothesis that has otherwise been unexamined.
>
>
> > [W3] ... investigate whether these are the only factors that contribute to representational similarity ... other factors (architecture, model size, training duration)...
>
> Thank you for the suggestion, we agree that these would be interesting experiments. To this purpose, we completed additional ablation (see Figures 19-23) on tuning duration, architecture, and model size.
>
> To summarise, we find that when the two models in a pair have different sizes/architectures, the effect of dataset overlap is diminished, whereas task overlap remains a strong driver of similarity; when tuning vision models for longer, there is a stronger correlation between dataset overlap and similarity, whereas the effects of task overlap is slightly diminished when tuning duration increases. Please refer to the discussion in section A.2.4 for details.
>
>
> > [Q1] How were the representations exactly extracted for the different models?
>
> The representations are extracted as follows.
> 1. ResNet/ViT: the inputs to the final fully-connected layer as used as features;
> 2. nanoGPT and Llama: the last hidden state as out by the transformer blocks are the features (we average over the context dimension);
> 3. Diffusion UNet: we use the outputs of `conv_act` as features as found in the diffusers' `UNet2DModel` class.
>
> We include the above information in section A.1.1.
>
>
> > [Q2] Why didn’t you use the CKNNA ... downsides of CKA in the Limitations section?
>
> We do use CKNNA, as well a few other metrics! **Please see the appendix for detailed results on all seven metrics we use.**
>
>
> CKA is a somewhat reliable metric [4] despite of its weaknesses, and the authors of CKA [5] demonstrated its many reliable properties (e.g., CKA with a linear kernel satisfies desirable invariances to orthogonal transformations and scaling). However, [3] pointed out in section A the limitations of CKA, which "has a very strict definition of alignment". Therefore, authors in [3] develop CKNNA which is a modified version of CKA as well as other metrics. We decided to present CKA in the main body because it is widely used (developed in 2019).
>
> Our goal is to demonstrate how dataset/task overlap may affect representational similarity and not to pick the best metric for our experiments; since different metrics each have their strengths and weaknesses, we decided to use a wide range of metrics found in [3] and present CKA in the main body.
>
>
>
>
>
>
> [1] Objective drives the consistency of representational similarity across datasets, ICML 2025
>
> [2] What shapes feature representations? Exploring datasets, architectures, and training, NeurIPS 2020
>
> [3] The platonic representation hypothesis, ICML 2024
>
> [4] Reliability of CKA as a Similarity Measure in Deep Learning, ICLR 2023
>
> [5] Similarity of Neural Network Representations Revisited, ICML 2019
>
>
>
>
> We thank the reviewer for their time. Please feel free to respond with any questions/requests for clarifications.

---

### Official Review · Reviewer_poVZ · 2025-11-11

**Soundness:** 3
**Presentation:** 3
**Contribution:** 2
**Rating:** 4
**Confidence:** 4

**Summary:**

This paper examines the extent that dataset and task overlap correlate with representation similarity. They use a new image dataset that fixes the images and varies labels and objectives; this isolates task overlap. They also perform some experiments on the correlation between jailbreaks and representation similarity.

**Strengths:**

- Novel causal interventions on causes of model representation alignment. The paper systematically modifiies dataset overlap and task overlap, and measures how these correlate CKA/other alignment metrics for both ViTs and small LLMs.
- Isolate task vs data overlap in representation similarities. Perform experiments on a number of different measures (CKA, nearest-neighbor scores, mutual KNN).

**Weaknesses:**

- Concerns about generality. In general, the results on language modeling seem to be in tension with the stated takeaways of the paper, and the authors do not properly explain this discrepancy. Namely, they note that for the Llama fine-tuning experiments, the models in general have low correlation between their CKA scores and dataset overlap. This is comparatively true for task overlap as well. This lower correlation is hypothesized to be due to the fact that the LLMs were fine-tuned, so a lower proportion of their total training tokens are explained by the new task/dataset training. The authors note in the main body that they also run nanoGPT from scratch; however, these results seem to only be discussed/provided in the appendix (e.g. Figure 13). While these results are not discussed in the paper and the actual correlations between overlap and similarity are not provided, it visually it appears in Figure 13/14 that the correlation between dataset/tasks is pretty weak. The authors should both provide these correlations address this discrepancy. This also goes for the jailbreak correlation results in section 6.
- This paper has some conceptual overlap with [1, 2, 3]. These are discussed briefly in the paper at the moment, but it would be great to see a bit more discussion here.


[1] Ciernik, Laure, et al. "Objective drives the consistency of representational similarity across datasets." Forty-second International Conference on Machine Learning.

[2] Klabunde, Max, et al. "Towards measuring representational similarity of large language models." arXiv preprint arXiv:2312.02730 (2023).

[3] Brown, Davis, et al. "Wild comparisons: A study of how representation similarity changes when input data is drawn from a shifted distribution." ICLR 2024 Workshop on Representational Alignment. 2024.

**Questions:**

See weaknesses.

---

> ### Author Response · Authors · 2025-11-23
> **Initial response (1)**
>
> We thank the reviewer for the review. We are glad that the reviewer found our contributions novel. In the updated PDF (**all updates are in blue** and longer new content is noted with text like "new ablation" or "new discussion" in blue), we
>
> 1. provide additional ablation on model size, architecture, and tuning duration in section A.2.4; specifically:
>     - Figure 19 investigates the effects of tuning duration,
>     - Figure 20 explores how model size ratio between pairs of models affects representational similarity at a fixed overlap,
>     - Figures 21 and 22 fix the model pairing (ResNet18+ResNet152 in Figure 21, ResNet50+ViT/Small in Figure 22) while varying the overlap;
> 2. modify our claims in the conclusion, introduction, and abstract to correspond more closely to our results;
> 3. add previously-missing trend lines to plots in the appendix;
> 4. include additional discussion of related works in section A.5;
> 5. discuss nanoGPT results in more detail (section A.2.2);
> 6. present full-model tuning (Figure 24) of Llama3.2-1B-Instruct in addition to LoRA tuning.
>
>
> We respond to the reviewer's questions and concerns below.
>
>
> > [W1a] In general, the results on language modeling seem to be in tension with the stated takeaways of the paper
>
> Thank you for pointing this out. We have modified the abstract, introduction, experimental results, and discussion to reflect the weaker LLM results. We would like to point out that the tuned Llama models exhibit moderate correlation between task overlap and representational similarity (see Figures 5, 24). However, we concede that the transfer jailbreaking for Llama and dataset overlap results for Llama and nanoGPT indicate that dataset overlap is not a strong driver for representational similarity in language models.
>
>
>
> > [W1b] and the authors do not properly explain this discrepancy.
>
> We agree that we should've considered the inconclusiveness of language generation results more thoroughly, in addition to providing "finetuning" as a reason. We give a more in-depth treatment here.
>
> We hypothesise that language results are less convincing partly due to the following: language datasets are larger and contain more general information, leading to models being more general after being trained on split datasets.
>
> A larger dataset causes each split to be more general, meaning that it is more likely for each dataset to contain enough varied data to train models that are better at generalising. We used TinyStories and Shakespeare for dataset overlap on nanoGPT, and both contain around 2 million tokens. In contrast, the largest classical image dataset we use is TinyImageNet, containing 100k images, with TinyStories containing 20 times more tokens than TinyImageNet has images. We admit this is a simplistic way of measuring the information contained within a dataset.
>
> We note that the above hypothesis is supported by diffusion model results: they exhibit weak correlation between overlap and representational similarity. Although we trained each diffusion model on CIFAR-10, the score matching training procedure increases the number of unique inputs $x_t\sim N(x_0,t^2 I)$ beyond the 50k images present in CIFAR-10 due to adding random Gaussian noise to all images. Hence, this provides evidence that training with more information reduces the effect of overlap on similarity.
>
> The reviewer might wonder if the results in Figure 19 (finetuning duration ablation for ResNet-50 on CIFAR-100) contradict the above hypothesis; it does not. Since we repeat over many epochs, there is no new data introduced.

---

> > ### Author Response · Authors · 2025-11-23
> > **Initial response (2)**
> >
> > > [W1c]  The authors note in the main body that they also run nanoGPT from scratch; however, these results seem to only be discussed/provided in the appendix (e.g. Figure 13). While these results are not discussed in the paper and the actual correlations between overlap and similarity are not provided, it visually it appears in Figure 13/14 that the correlation between dataset/tasks is pretty weak. The authors should both provide these correlations address this discrepancy. This also goes for the jailbreak correlation results in section 6.
> >
> > We have provided additional discussion concerning nanoGPT results in section A.2.2. As the reviewer notes, nanoGPT does indeed exhibit relatively weak correlation between overlap and similarity. Due to the page limit, we defer discussion concerning nanoGPT to the appendix since results (even if weak) on larger language models elicit more interest from reviewers. Please see our response to [W1b] for a general comment on the discrepancy between strong vision results and weak language results.
> >
> > We update our figures with linear trends and $R^2$ values for nanoGPT and ResNet-18 results in the appendix (Figures 9, 10, 13, 14). For jailbreaking, we provide trend lines and $R^2$ values in Figure 7. Generally, for transfer jailbreaking attacks, the $R^2$ values are very close to 0 since there is almost no relationship between overlap and transfer attack success rate.
> >
> >
> >
> > > [W2] This paper has some conceptual overlap with [1, 2, 3]. These are discussed briefly in the paper at the moment, but it would be great to see a bit more discussion here.
> >
> > Thank you for the suggestion, specifically of [3] since we did not cite [3] in our original submission. We have included additional discussion in the last paragraph of section A.5. The reviewer may find the entire section A.5 to be useful, since we discuss additional related works beyond the three suggested by the reviewer.
> >
> >
> >
> >
> > We thank the reviewer for their time. Please feel free to respond with any questions/requests for clarifications.

---

### Author Response · Authors · 2025-11-23

We realised that Table 3 (mutual information) contained an error in the original submission; this error has already been corrected in the updated submission. Specifically, we swapped the column headers for Task + Dataset Overlap and Task Overlap (second-last and last columns). We checked the original plotting scripts on GitHub (timestamped prior to submission), and the reason for this error is due to accidentally swapping the output order of the three overlap types in the script we used to generate the table; unfortunately, we forgot to swap the column headers to reflect the changes in script. This does not affect our central conclusion that both dataset and task overlap drive representational similarity. We have modified the submission's discussion to reflect the updated result.

---

### Author Response · Authors · 2025-12-04
**Rebuttal Summary**

We appreciate the reviewers' and AC's time and careful evaluation of our submission. We summarise the paper and the rebuttal process to help the AC in making their recommendation.

## Summary of Contributions
Our paper provides the first systematic empirical study of *causal drivers* of representational similarity in neural networks. We isolate and independently manipulate two training-time factors, namely **dataset overlap** and **task overlap**, and measure their effects on similarity across multiple modalities (image classification, image generation, and language generation). To facilitate controlled experiments, we introduce the `ColorShapeDigit800K` dataset which allows us to control dataset overlap and task overlap independently and quantify task overlap without confounding factors. We further demonstrate downstream effect of representational similarity: **more similar vision models exhibit greater vulnerability to transfer attacks**.

## Argument for this work's novelty
Reviewers find our paper "creative" [k4Mo], "clearly motivated" [k4Mo], "well-designed" [k4Mo,dQv7], "enjoyable to read" [k4Mo], "novel" [k4Mo,poVz], and that our paper "studies an interesting question" (NmnH).

Highlights of our work's novelty & contributions:

1. **Causal isolation of dataset vs. task overlap** as causes of representational similarity, which has not been explored in prior works. Existing works measure similarity across pretrained models; none manipulate overlap between  datasets to quantify their effects on similarity.
2. **Quantifiable and independent task-overlap control**, enabled by `ColorShapeDigit800K`. This allows us to show that task overlap alone—holding data constant—drives representational alignment.
3. **Thorough and systematic experiments** on how overlap causes representational similarity, extending beyond image classification models to diffusion models and (large) language models. While language-model effects are weaker, we discuss this discrepancy in the rebuttal.

Reviewers repeatedly highlighted our controlled design, breadth of evaluation, and clear isolation of factors long assumed (*but never demonstrated*) to influence representational similarity.

## How Reviewer Concerns Were Addressed
We fully addressed all reviewer comments during rebuttal:

- We **clarified limitations for language models**, revised claims throughout the paper, and added more thorough analysis, including full-model tuning (Fig 24) and additional discussion of language modelling results. In short, we suspect the more general nature of language datasets cause models to be more general after tuning on overlapping datasets, leading to weaker trends between overlap and similarity (see our response to reviewer [k4Mo]'s [W1a] for extended discussion).
- We **provided additional ablations** for dataset overlap and task overlap method (1) on model size, architecture differences, and tuning duration (see Fig 19-22 in the updated PDF).
- We **expanded related-work discussion**, including deeper treatment of overlaps with prior papers suggested by reviewers [poVZ,dQv7] (see section A.5 in the updated PDF).
- We **added technical details**, including representation extraction procedures, pretrained weights, and explanations of similarity metrics (see text in blue in the updated PDF).
- We **ran additional experiments**, including varying dataset overlap for `ColorShapeDigit800K` to demonstrate robustness of task-overlap effects (see Figure 29 in the updated PDF).
- We **corrected and clarified tables/figures** flagged by reviewers and updated discussions to align conclusions more closely with empirical results.

We believe our updated PDF together with our rebuttal **fully address all concrete concerns** and provides a clear, novel, and substantively improved submission. Additionally, in our rebuttal we provide convincing arguments for the novelty of our work.

---

### Meta-Review · Area_Chair_fAkd · 2026-01-07

**Summary:**

The paper investigates the factors that cause model representation similarity with two separate factors: dataset overlap and task overlap. The reviewers acknowledge that the paper handles an interesting and important question. Also, the paper's well-written manuscript and experiment design are noted as major strengths. However, the reviewers [poVZ, NmnH, dQv7] raise concerns about the generalizability of experiments to practical models. While `ColorShapeDigit800k` is an interesting synthetic dataset, it differs significantly from practical tasks, such as LLM [poVZ], SSL objectives [NmnH], and real-world models [dQv7].

As three of the four reviewers raise concerns about generality, I agree that it is a significant weakness of the paper. There is no problem with utilizing a synthetic task and dataset, but it should be sufficiently related to real-world applications to connect findings to practical cases and make a contribution. Overall, the reviewers' opinion leans toward rejection, and I can't recommend acceptance for this paper.

I carefully reviewed the Reviewer dQv7's review and don't think it's problematic. `Single similarity metric (CKA)` is just an additional claim, and the generality issue still holds without this part. So, it will not be considered as an AI-generated review in this meta-review.

**Reviewer Concerns:**

Remaining concerns
- Reviewer poVZ
  - Concern about generality
  - Conceptual overlaps with prior works
- Reviewer NmnH
  - Task complexity is too simple to be generalized to actual SSL objectives
  - Unclear insights behind experiments
- Reviewer k4Mo
  - Unfair comparison with vision & language model
- Reviewer dQv7
  - Limited novelty: overlaps with prior works
  - Limited generalizability to practical real-world models

**Reviewer Scores:**

- Reviewer poVZ: Would maintain the current score.
- Reviewer NmnH: Would maintain the current score.
- Reviewer k4Mo: Would maintain the current score.
- Reviewer dQv7: Would maintain the current score.

---

### Decision · Program_Chairs · 2026-01-26

Reject